# Reducing the Cooling Loads of Buildings Using Shading Devices: A Case Study in Darwin

Aiman Mohammed [1], Muhammad Atiq Ur Rehman Tariq [2,3], Anne Wai Man Ng [4,5], Zeeshan Zaheer [4], Safwan Sadeq [6], Mahmood Mohammed [7] and Hooman Mehdizadeh-Rad [4,5,*]

[1] Faculty of Civil Engineering and Built Environment, University Tun Hussein Onn Malaysia, Parit Raja 86400, Malaysia; aimanfahd195@gmail.com

[2] College of Engineering and Science, Victoria University, Melbourne, VIC 8001, Australia; atiq.tariq@yahoo.com

[3] Institute for Sustainable Industries & Liveable Cities, Victoria University, Melbourne, VIC 8001, Australia

[4] College of Engineering, IT & Environment, Charles Darwin University, Darwin, NT 0810, Australia; anne.ng@cdu.edu.au (A.W.M.N.); eng.zeeshanzaheer@gmail.com (Z.Z.)

[5] Energy and Resources Institute, Charles Darwin University, Darwin, NT 0810, Australia

[6] Department of Mechanical Engineering, University Teknologi PETRONAS Malaysia, Seri Iskandar 31750, Malaysia; safwan_19001798@utp.edu.my

[7] Faculty of Engineering, University Putra Malaysia, Seri Kembangan 43400, Malaysia; gs61624@student.upm.edu.my

* Correspondence: hooman.mehdizadehrad@cdu.edu.au

**Abstract:** It is estimated that almost 40% of the world's energy is consumed by buildings' heating, ventilation, and air conditioning systems. This consumption increases by 3% every year and will reach 70% by 2050 due to rapid urbanisation and population growth. In Darwin, building energy consumption is even higher and accounts for up to 55% due to the hot and humid weather conditions. Singapore has the same weather conditions but less energy consumption, with only 38% compared to Darwin. Solar radiation can be defined as electromagnetic radiation emitted by the Sun and the Darwin area receives a large amount of solar radiation; building energy consumption can be reduced hugely if this radiation is blocked effectively by analysing appropriate shading devices. This study investigated the influence of different types of shading devices on the cooling load of a town hall building located in Darwin, Australia, and proposed the optimal shading device. The results showed that the horizontal fins led to a 5% reduction in the cooling load of the building. In contrast, adding a variation to the device angles and length increased the savings to 8%. The results demonstrated that the overhangs were more efficient than the fins, contributing 9.2% energy savings, and the cooling reduction savings were increased to 15.5% with design and length variations.

**Keywords:** building energy modeling (Revit); building energy efficiency; cooling load; solar shading devices

## 1. Introduction

In today's world, around 40% to 50% of electricity is consumed by buildings [1,2], with 30% consumed by HVAC systems [3]. The World Health Organization (WHO) indicated that "cities will house 50% of the world's population by 2050, rising to 70% by 2050. This urbanization and population growth will result in an increase in building energy consumption" [4,5]. Many elements can influence a building's performance or energy demand, and if managed properly, energy demand can be significantly lowered. Building components and envelopes are to blame for the high cooling energy demand. Occupant behavior, operating hours, and the number of appliances inside the structure can all raise the amount of energy consumed [6].

Windows that are capable of blocking excessive radiation can improve the cooling load of a building. In contrast, windows and other façades with high transmission values

can increase the cooling load [7]. However, these are not the only factors when it comes to heat gained from solar radiation. The Sun angle, time, and location also play a crucial role in emitting radiation [8]. The cooling load of a building air-conditioned space can be divided into two main categories: internal heat gain and external heat gain. Internal heat gain originates from the occupants, electric lighting, computers, and other equipment [9]. In a climate zone such as Victoria, with a hot summer and cold winter [10], the summer sun is high and lower in winter. However, these windows also present the opportunity to utilise the natural light indoors, precisely eliminating the artificial source of light and saving building energy loads. Many authors suggest that natural light inside buildings during the day improves residents' health and visual comfort [11].

Allowing the required amount of light into buildings and eliminating excessive lighting is tricky and requires a plan to install shading or tinting windows to a certain amount. The extra glare of sunlight can be controlled by applying internal, external shading, or electrochromic technology. Studies were done to measure the glare index and daylight factor in a simulation program design-builder and suggested the optimal level of light in a building [12,13].

In reality, lighting quality is not directly measurable but is an emergent state created by the interplay of the lit environment and the person in that environment. Veitch et al. [14], who investigated the determinants of lighting quality, mentioned that "one cannot measure quality in the same sense as one measures length, mass, or lumen output, and lighting quality can only be assessed using behavioral measures." Consequently, it cannot be measured directly, as indicated by [14]; however, as mentioned, it was estimated using behavioral measures. It is emphasised that this method is a simple, practical method since no real person was involved in the study.

Depending on the panel setup, neighboring panels can cast shadows over lower panels in the same system. This issue typically only arises for in-ground installations. Panels can actually be shaded by the roof they are on. Depending on the sun's angle and the time of day, different parts of a roof (such as a chimney or dormer) can block sunlight to certain panels. Therefore, we cannot discuss shade without mentioning clouds. Despite the fact that clouds do technically block out the sun and cast shade, the clouds still let some sunlight through, which means solar panels still can produce energy, albeit at a lower efficiency. The shaded solar panels produce less power than those in direct sunlight. Exposure to less powerful sunlight is the obvious contributor to lowered efficiency, but the design of a solar installation, specifically, the panels and their inverter(s), also matters. This research constitutes a relatively new area that has emerged from this solar energy system that can be used at any time when the sun is shining; however, more electrical power can be expected when the sun is very bright and shines directly on the PV module. Shading is one of the aspects that can have an impact on PV systems. Many academics have studied the usage of bypass diodes in shading scenarios, but the fact is that shading must be thoroughly analysed and avoided since it can lead to a breakdown of the shaded module.

This research aimed to assess different technologies used in commercial buildings in hot and humid climate conditions to reduce the cooling loads and keep the comfort level at an optimal level. Additionally, this study focused on passive cooling strategies via different shading devices by considering the state of the building to help reduce the heat gain of the building due to solar radiation. Furthermore, this study investigated and compared various shading devices that are available and compared the depth effect of these shading devices using building energy modeling software (Revit). The results were then compared with the simulation model to suggest the best design shading device.

## 2. Literature Review

Shading devices are part of the solar control façade systems defined in the same standards, and their installation has become mandatory for some public buildings as a result of the 2015 revisions to the regulations. A common shading device is one that generates a pleasant indoor environment by appropriately regulating or blocking incoming

solar radiation, thereby reducing the cooling or heating load of the room and selectively allowing natural lighting and vistas. Although sunshine through window glass helps to reduce heating demands in the winter, it can create a large rise in cooling loads in the summer due to indoor heat gain from solar radiation. As a result, by using shade devices and supplementing the weak parts of windows in the summer, it is feasible to cut energy consumption while still creating a comfortable indoor environment [15].

Several studies on shading devices have previously been undertaken, and the existing literature can be summarised as follows: Al-Tamimi and Fadzil [16] investigated the feasibility of using shade devices to reduce the temperature of tropical high-rise residential structures. They used simulations to examine ideal external shading devices that can minimise incoming heat and hence improve energy usage, with a focus on Malaysia's hot and humid climate and the internal temperature control effect for high-rise residential structures. Kim et al. [17] conducted an energy simulation utilising a computer model designed for Korean residential structures based on practicality in order to introduce ideal external shading devices via comparative research on the thermal performances of residential building external shading devices. Palmero-Marrero and Oliveira [18] studied the effect of a louvred sunshade system, evaluated the performance of shading devices based on orientations and conditions, and analysed the effects of the louvred sunshade system that change depending on a variety of parameters. Kim et al. [19] investigated the cooling and heating energy consumption of Korean office buildings when horizontal shading devices or Venetian blinds were utilised, as well as optimal shading devices based on areas and orientations. Lee et al. [20] conducted a climate index development study utilising local weather data in order to understand the features of the local climate in the early design stage and confirm the validity of shading devices that can be judged by the user. Kim et al. [21] investigated cooling load reductions by analysing the reduction of cooling loads in office buildings with a high cooling load in order to confirm the effect of an effective shading device design on office buildings.

A. Gagliano et al. [22] investigated the cooling load of a lecture hall in the hot, humid climate of Malaysia. They used insulation materials PASB (polyethylene aluminum single bubble) on the external wall and simulated the insulation material with CFD software using collected data for a one-year duration. A reduction of 3 °C was achieved using the insulation materials, resulting in a lower cooling load requirement. A. Gustavsen et al. [23] also installed polyurethane on the outer side of a house wall and achieved a 28% reduction in the house's cooling load. Another study was undertaken in Hong Kong by J.-W. Lee et al. [24], which involved applying polystyrene on the external and internal walls of the building; they achieved a 38% reduction in cooling load.

These experiments support the idea that installing low-U-value materials on the wall helps to minimise the cooling load of buildings. The contribution from windows is considered to be one of the most effective factors of heat gain/loss in buildings. Z. Yang et al. [25] claimed that up to 60% of building energy loss is due to windows with a 30% window to wall ratio (WWR) of a two-story building. Moreover, by decreasing the WWR to 20%, the energy loss was 45%. Other factors of window heat loss are the thermal conductance of window material; a better insulating window with a minimum U-value can significantly reduce these losses [26].

Subsequently, it was found that double-glazed windows are 50% more efficient than single-pane windows and have a very long life [27]. Though this technology is getting more common daily, many modern versions are being designed and evaluated for their performances. Among the current versions is aerogel fitted, vacuum, and PCMs fitted glazing. Aerogel is a world-class insulating material that is employed in the space industry due to its extreme insulation properties and delicate nature. It is a costly material, but scientists are trying to reduce its manufacturing costs. Aerogel is placed between two layers of glass; with it being very lightweight, the increase in mass of the window structure is a minor concern. According to C. Buratti and E. Moretti [28], aerogels are available in various transparencies, ranging from fully transparent to translucent to opaque, with

variations in their costs. The aerogels were tested; it was found that their heat transfer coefficient is extremely low, having a value of 0.013 W/m$^2$K [29]. Since aerogels are available in two types, known as monolithic and granular, an investigation was done by J. L. Aguilar-Santana et al. [30] and C. Buratti and E. Moretti [28] to compare both types. It was concluded that monolithic is much better in terms of its solar transmittance in the form of light and insulation ability, with an overall U-value of 0.60 W/m$^2$K.

The purpose of this study was to confirm that shading devices can be implemented to improve the visual comfort of indoor building occupants by filtering excessive sunshine while allowing appropriate daylight to enter through windows. Tzempelikos and Athienitis [31] investigated the control of shading devices for building cooling and illumination control and offered advice on shading device performance and window glass ratio design. They conducted investigations in order to provide instructions on thermal properties, shading control, and how to choose the glass ratio of the façade. Choi et al. [32] conducted research on a parametric louvre design system for optimising the shape of the louvre and established a parameter design methodology that integrates heat and the design pyramid by doing a thermal study and investigating a parameter design methodology. Karlsen et al. [33] created a sun-shading control approach for Venetian blinds used in cold climate office buildings to ensure acceptable energy use and indoor environmental performance.

Khoroshiltseva et al. [34] developed a multiobjective evolutionary design technique for optimising shading devices included in refurbishment kits for an existing residential structure in Madrid. Singh et al. [35] evaluated the effect of increased shadow transmittance values on the energy and visual performance of an office building. The research was carried out at Shillong, which is characteristic of chilly climates in India. A variety of glass and internal roller shade combinations were simulated for south-, west-, north-, and east-facing offices with varied window sizes, glazing qualities, and shading methods. Eom et al. [36] discriminated between periods when shade devices are required and periods when they are not required by calculating the balance point temperature using simulations, and they constructed shading devices based on the periods split in this way. As a result, they offered a specification for ideal shading devices within the size range of shading devices specified by solar altitude, as well as a quantitative basis for projection length based on annual heating and cooling demands. Kim et al. [37] used IES 5.5.1 to analyse annual cooling and heating loads, as well as the amount of sunlight on the living room floor surface, and assessed the effects of movable horizontal shading devices to assess the impact of a new type of movable horizontal shading device on the indoor thermal environment and solar access performance.

Kim et al. [38] used the e-Quest program to evaluate the shading coefficient applied to energy-saving building envelope technology of office buildings and the loads of different types of horizontal shading devices according to orientations, and analysed the envelope elements according to orientations. Kim and Yoon [39] conducted a quantitative assessment of the various façades by taking into account the physical properties of the envelope components that can be selected in the envelope design, calculating annual loads through simulation, particularly with respect to the combination of windows and fixed external shading devices, and analysing the design suitability. Using a building energy analysis tool, Song et al. [40] investigated the complete solar irradiation of the vertical glass surface dependent on the length of the horizontal shading device according to the orientations affecting the perimeter boundary in office buildings in Seoul. Kim [41] conducted a study to derive improvement methods of solar radiation control standards of windows and shading devices based on an analysis of our countries and other countries' related standards by analysing the current status of major countries' energy-saving design standards of buildings and performing a comparative analysis of them with the national standards and investigating complementary elements for the national standards and necessary amendments.

Shading systems are created as part of the building to prevent unwanted daylight that would cause high internal temperatures and unwanted lighting, as well as to reduce the

additional operating expenses of the building system. Such as shading system strengthens the shading system of the building and establishes the design capabilities in order to adapt to the user [42].

Shading devices are classed as interior or outdoor shading devices based on where they are installed. Venetian blinds and roll screens are examples of interior shading devices, while louvres, light shelves, and awnings are examples of external shading devices [43]. Furthermore, shading devices can be characterised as fixed, manual, or movable based on how they operate [44]. Shade devices come in a variety of materials, sizes, and shapes, and can be installed in a variety of locations within a building, such as windows or as part of the architecture [45].

Because they move in line with the direction of the sun, movable awnings perform better in terms of daylight control in different seasons [46]. Fixed shading devices, on the other hand, are more suitable for implementation, particularly in Iran, where there are economic concerns and the employer accepts a more economical plan, and it also costs less to implement, in addition to being easier to design and implement.

The height of the sun and the azimuth when the shadow is required decide the fixed shading design [47]. On the one hand, horizontal shadows have the largest effect on the south side, and the length of the ridge is determined by taking into account the sun's altitude angle. Azimuth, on the other hand, is a significant consideration for vertical shading that is utilised on the east or west side of a building where the solar height is low. It is effective to insert vertical fins at small distances to boost the protection speed while shortening the protrusion length. Regardless of the orientation, eggcrate-shaped shadows merge vertically and horizontally, taking into consideration both solar and azimuthal heights. However, one downside is that the function of natural light is lost as a result of overprotection [48].

However, in the current literature, most studies on façade optimisation only focus on a single façade orientation, which is the equator-facing side, because the observed buildings are typically located in high-latitude regions, e.g., [49,50]. Meanwhile, in tropical regions, optimisation of all façade orientations is required due to the relatively high solar radiation and long sunshine duration. Although some attempts have been made in the prior work to address this issue, most of these studies focus on particular case studies, such as the design of shading devices in tropical office buildings according to recent studies conducted in [51,52], and ignore other tropical regions with specific case studies, for instance, Casuarina Darwin, which might provide different results, as expected by the authors of this study. Moreover, this study provides a review of recent techniques and presents a comparison against earlier methods; it was found that some studies focused on fixed (or static) shading devices [53,54], while some focused on adaptive (or any of the alternative adjectives) shading devices [55,56]. It was concluded that there is still a lack of studies on adaptive shading devices on business buildings façades in the tropical context [57]. In addition to the main conclusion of the review [25], it is also mentioned that fixed shading devices have been a popular choice for application in the tropics, particularly to optimise thermal and daylight performance, despite the limitation in responding to varying weather conditions. Meanwhile, adaptive shading devices are less popular than fixed devices, mainly due to the complexity and the attributed operational costs [58]. Based on the above critical review, it was strongly recommended to adopt the optimised shading devices among other possibilities as a compromise.

## 3. Materials and Methods

The primary purpose of this study was to investigate the influence of different types of shading devices on the cooling load of a town hall building located in Darwin, Australia, and propose the optimal shading device. For this reason, the research method involved simulation modeling as shown in Figure 1. In this way, at first, the selected building was modeled in the Revit software and the energy was analysed in terms of daylight and heat. The tool has the capabilities to model and performs simulations for different types

of buildings. Besides that, it can also improve the existing building technologies with emissions and energy use by buildings. It describes the different cooling load factors in components and external heat gain by the building. Additionally, it gives an overview of solar shading devices and explains how the sun path can affect the performance of a shading device. Visual comfort, thermal comfort, building energy rating system in Australia and Darwin weather conditions are explained. Based on an extensively conducted critical review, a systematic methodology was developed to analyse the performance of the different types of shading devices.

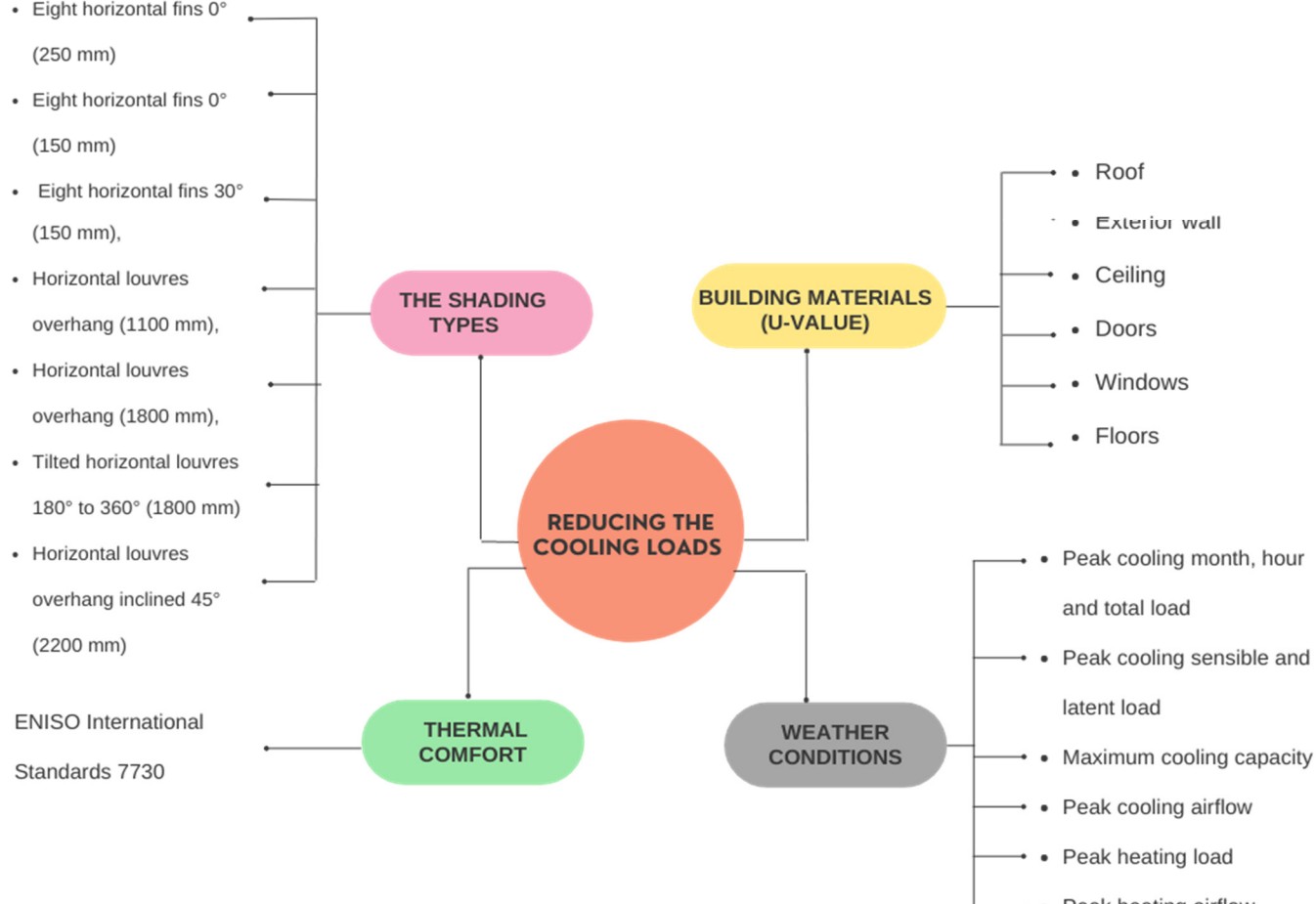

**Figure 1.** Methodology adopted for reducing the cooling loads in the building.

The authors selected Casuarina Darwin because this building is considered to be in the middle of the city, which will give more reasonable data for our study. Besides that, the effect of different shading types applied to the exterior of the building was analysed and their impact on the cooling load requirements of the building was studied. Casuarina Darwin is a multipurpose town hall that is used for public events in Darwin, Australia. Furthermore, this building is located in a tropical region that features high temperatures, which serves the primary purpose of this study. The real-time sun path of this building was also generated through Suncalc, as shown in Figure 2.

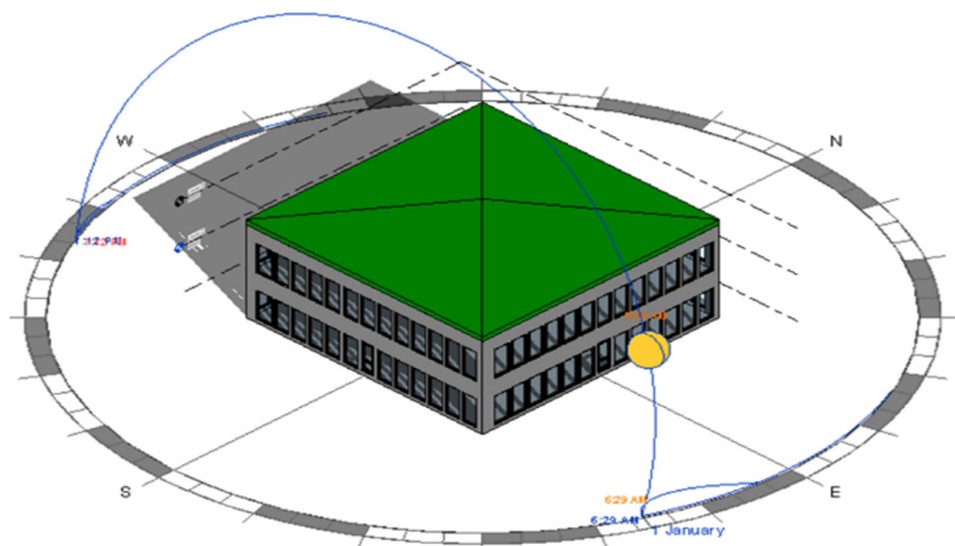

**Figure 2.** Three-dimensional simulation by Autodesk Revit software that provides a view of Casuarina Darwin with the sun path.

The altitude angle is the angle of the sun above the horizon, achieving its maximum on a given day at solar noon. The azimuth angle, also known as the bearing angle, is the angle of the sun's projection onto the ground plane relative to the south.

Shading devices can have a dramatic impact on building appearance. This impact can be for better or worse. The earlier in the design process that shading devices are considered, the more likely they are to be attractive and well-integrated in the overall architecture of a project.

The highest sun angle measured from the ground was 72.66° at 12:44 p.m. in December 2021 for the summer azimuth on Darwin. The lowest sun angle measured was recorded as 22.420° at 12:48 on 21 June 2021 for the winter azimuth from the four different directions of the building. These are important because they play a vital role in determining the cooling loads, as shown in Figure 2.

Revit is used to design, document, visualise, and deliver architecture, engineering, and construction projects. Revit is software that uses BIM technology encompassing a complete life-cycle of a project, including planning, designing, building, estimating, operating, and maintaining the facility. Revit is aimed at more complex projects of whole infrastructures, where Revit will recognise elements. As design software, it leaves much to be desired. However, its strengths lie in its handling of services. It has a very robust error-detection system that can quickly identify potential conflicts in the services plan of a project. It is possible to coordinate very easily with the services consultants if they are also working in Revit. Revit has changed architecture design, drafting, and modelling processes by accomplishing substantial improvements in precision and efficiency. With Revit software, architectural designers can now rapidly sketch out rough layouts of a floor plan or make changes to the standard set of building designs and instantly let their customers preview their future homes [59,60].

The changes in components are in terms of windows and shading devices because they are responsible for most of the cooling load in Darwin. Another software can be utilised to conduct these load calculations called HAP (Hourly Analysis Program). This software generates a more accurate result and is based on the ASHRAE standards [61].

The Hourly Analysis Program (HAP) is a component of the program libraries, which is one of the major research companies in the market. HAP licenses and reports outputs that are used internationally in the field, as well as calculates the hourly energy analysis by using 365 × 24 h of airflow information by predicting the heating and cooling loads of the building, the number of days per year, and 100% working condition to simulate the hourly airflow design and the operation of the selected mechanical system devices. The

HAP interface contains the meteorological information of specific centres in order to obtain the correct information according to the country of the building [62].

ASHRAE standards are entered into the program to calculate heating and cooling loads. Steps involved in calculations when using HAP are as follows: location information—where the location information is entered in the program; room description—the area and height of the spaces should be entered; the amount of airflow conditioned in the area is entered; the lighting, electrical equipment, and the heat gain loads from the room are entered; window and door areas in the same direction are entered; the roof area is entered; the floor features of the room are entered; infiltration information is entered; system selection—room or zone system selection; defining building components—profiles, wall properties, roof properties, window features, door features, and shadow properties are defined [63,64].

We selected shading because Darwin experiences an adequate amount of sunny and partly sunny days. Direct sunlight can be blocked or reduced before getting to the window, which is the weakest link for thermal losses occurring in a building.

To implement passive cooling technologies in buildings and reduce the cooling load, research on several passive cooling designs was undertaken around the globe. Furthermore, a study based on building information modelling software was undertaken. This approach of software-based research focused on the information available about shading devices, windows, and other building materials in published papers and experiments. Many studies were undertaken with the Revit and were considered the primary source of data. It is an intelligent 3D modelling process that provides integrated, comprehensive engineering and architectural tools to help architects, engineers, and designers enhance their understanding of the energy performance of the buildings at particular locations and parameters. In our case, we modelled the building design and conducted a basic load calculation for the building. These load calculations were undertaken for the essential materials used in building components and considered the baseload calculation for other installed or improved features [65].

### 3.1. Simulation Process

The simulation for the cooling load using Revit software was conducted in several steps (Figure 3) to ensure that reliable results could be achieved. This simulation was conducted for the basic original building design at the Casuarina Darwin, shown in Figure 4.

First step: This first step included modelling the building consisting of the floor and elevation. In this step, the total building air-conditioned area was identified as a single-zone system, and the variable air volume duct system was assumed during this phase.

Second step: The building orientation, location, weather data, and date were considered.

Third step: The building materials, such as window and doors types, were added from the library. These materials were set as constant for all the simulations done throughout the experiment to avoid material change on the load calculations.

Fourth step: In this step, the building's cooling load simulation was obtained without any shading device as a basic load calculation.

Fifth step: After designing the building, we designed different types of shading devices that varied in lengths and angles and applied them to the designed building separately.

Sixth step (final): The different shading effects were applied to the building. The cooling load simulation was conducted again to observe any variation in the cooling loads.

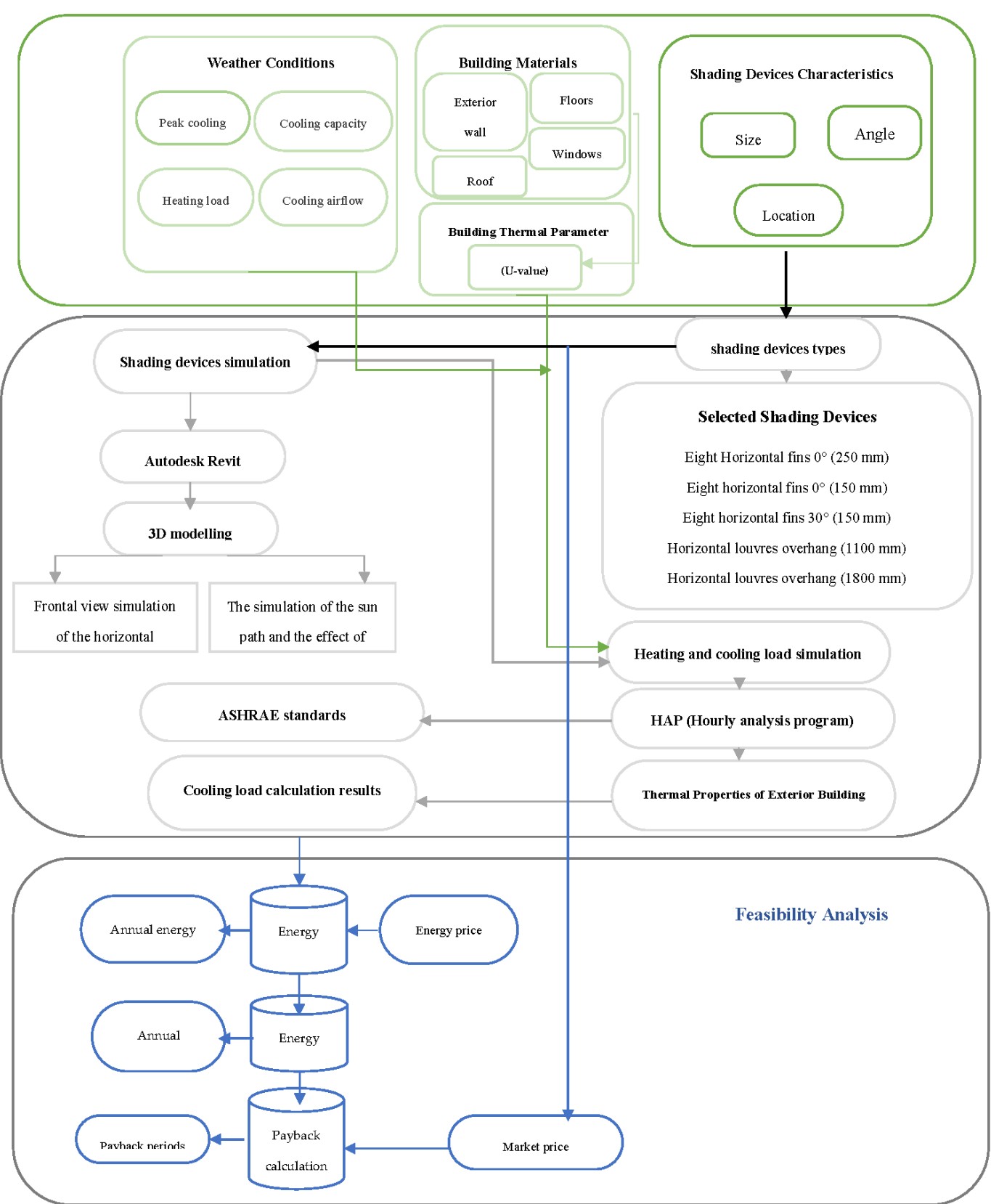

**Figure 3.** Methodology adopted for the feasibility calculations for shading devices in this research.

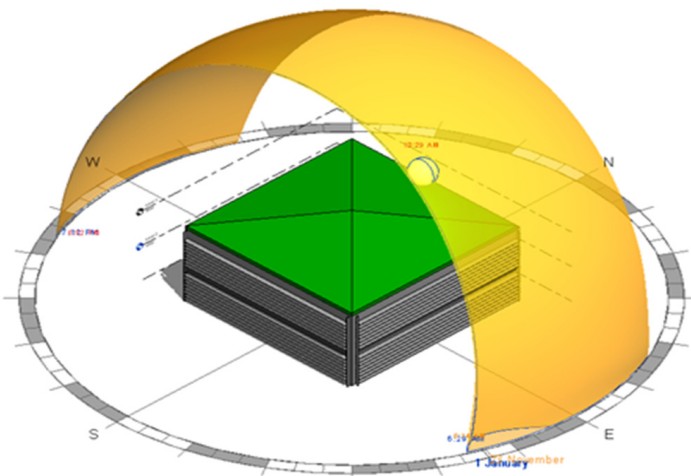

**Figure 4.** The simulation of the sun path and the effect of the applied 250 mm horizontal fins.

### 3.2. Assumptions

Some initial assumptions were made to help us with simplifying our problem statement. The assumptions are listed below:

- Internal loads, such as occupancy and electric appliances, stayed the same.
- The shades had uniform material properties irrespective of their orientation.
- All the shadings applied were of the same material and colour.
- No extra coatings were applied to the shadings.
- Uniform airflow was assumed under the applied shade.
- The building material properties remained the same in all calculations.
- The entire building was divided into two zones.
- The properties of the VAV duct system were assumed to perform the simulations.
- Any internal shades present were neglected.
- Orientation of the building was the same for all simulations.

### 3.3. Load Calculations for the Modelled Site

Modelling the building using Revit divided the building space into different spaces and zones. A total of two zones were created to allocate a selected area to a sensor, with one zone on each of the ground and first floor of the building. The space calculation was obtained after the building was divided into several zones. The load calculation was made based on the Darwin region. The following building parameters were obtained from the simulation, with a total floor area for both floors being 784 m$^2$ and a volume of 3077.51 m$^3$. Other factors are shown in Table 1.

**Table 1.** Details of Casuarina Darwin's project specification parameters.

| Building/Project Specification | |
| --- | --- |
| Building type | Town hall |
| Area (m$^2$) | 784 |
| Volume (m$^3$) | 3077.51 |
| Height (m) | 6 |
| Location | Casuarina Darwin, NT, AU |
| Latitude | $-12.46°$ |
| Longitude | $130.84°$ |
| Summer dry bulb | 30 °C |
| Summer wet bulb | 27 °C |
| Winter dry bulb | 20 °C |
| Mean daily average | 4 °C |

The building was designed using a selected heat transfer coefficient for the exterior components. These selected essential components stayed the same throughout the simulation experiment, except for window types and shading devices; the data were collected for the study location (Casuarina Darwin). The thermal properties of the exterior building materials are shown in Table 2.

**Table 2.** Thermal properties of the exterior building materials.

| Thermal Properties of Exterior Building Materials | |
| --- | --- |
| Parameter | U-Value (W/m$^2$ K) |
| Roof | 1.275 |
| Exterior wall | 0.810 |
| Ceiling | 1.361 |
| Doors | 6.870 |
| Windows | 5.692 |
| Floors | 2.958 |

### 3.4. Cooling Load Calculation Basic

The first simulation was taken from the site and the following results were obtained, as shown in Table 3. These results were based on the basic component coefficient with no shading effect.

**Table 3.** Basic cooling load calculation results and checksums.

| Basic Cooling Load Calculations | |
| --- | --- |
| Inputs | |
| Building Type | Town hall |
| Area (m$^2$) | 784 |
| Volume (m$^3$) | 3077.51 |
| Calculated Results | |
| Peak Cooling Total Load (W) | 175,151 |
| Peak Cooling Month and Hour | January 4:00 p.m. |
| Peak Cooling Sensible Load (W) | 130,012 |
| Peak Cooling Latent Load (W) | 45,140 |
| Maximum Cooling Capacity (W) | 169,621 |
| Peak Cooling Airflow (L/s) | 9466.4 |
| Peak Heating Load (W) | 2474 |
| Peak Heating Airflow (L/s) | 858.8 |
| Checksums | |
| Cooling Load Density (W/m$^2$) | 223.38 |
| Cooling Flow Density (L/(s m$^2$)) | 12.07 |
| Cooling Flow/Load (L/(s kW)) | 54.05 |
| Cooling Area/Load (m$^2$/kW) | 4.48 |
| Heating Load Density (W/m$^2$) | 3.16 |
| Heating Flow Density (L/(s m$^2$)) | 1.10 |

### 3.5. Cooling Load Calculation Basic

The method we chose for this research was based on software simulation because it was suggested by many research studies that this software can predict results with very high accuracy. Additionally, there is no setup available at the CDU laboratory to conduct on-site heat gain and cooling load experiments. Furthermore, investigations of shading devices, building materials, and windows are expensive and hard to maintain within a workshop. This kind of experimental setup will need an extensive form to calculate anything for a day or time of the year. However, fortunately, the university has provided the premium version of the simulation software Revit, where most simulation experiments can be undertaken.

### 3.6. Cooling Load with Shading Effect

The different shading types that were simulated were as follows:

a.   Eight horizontal fins 0° (250 mm);
b.   Eight horizontal fins 0° (150 mm);
c.   Eight horizontal fins 30° (150 mm);
d.   Horizontal louvres overhang (1100 mm);
e.   Horizontal louvres overhang (1800 mm);
f.   Tilted horizontal louvres 180° to 360° (1800 mm);
g.   Horizontal overhanging louvres inclined 45° (2200 mm).

The shading of eight horizontal fins was applied to each window and the cooling load was calculated. The width of the fin was 250 mm while it was at an angle of 0° to the horizontal. The sun path and shading are shown in Figures 4 and 5.

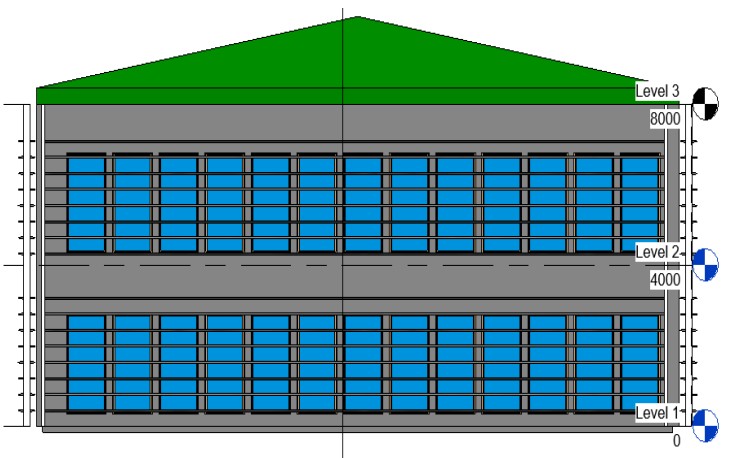

**Figure 5.** Frontal view simulation of the horizontal fins 250 mm applied.

Then, the shading of eight horizontal fins was applied to each window and the cooling load was calculated, where the width of the fin was 150 mm while it was at an angle of 0° to the horizontal. Figures 6 and 7 show the shading type applied to the building.

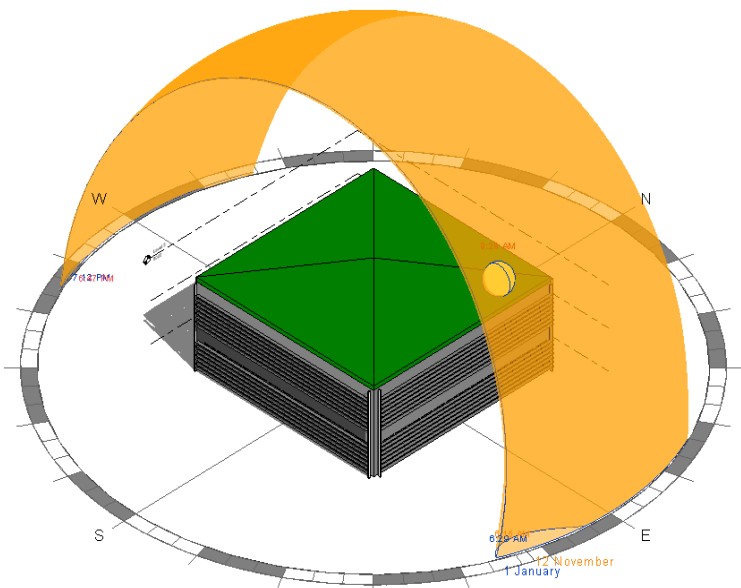

**Figure 6.** The simulation of the sun path and the effect of the applied 150 mm horizontal fins.

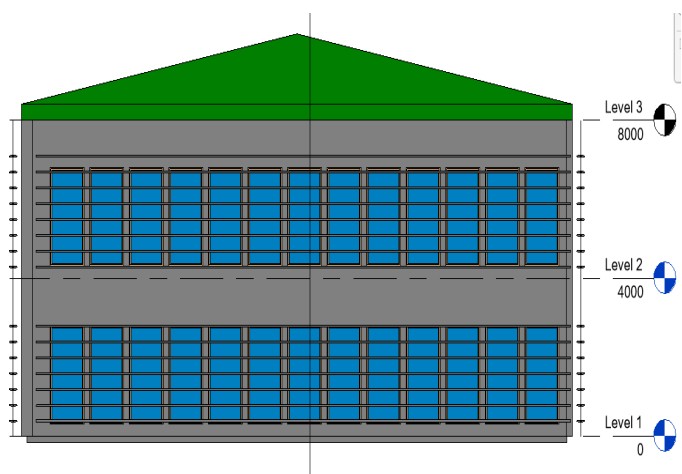

**Figure 7.** Frontal view simulation of the applied 150 mm horizontal fins.

Then, the shading of eight horizontal fins was applied to each window and the cooling load was calculated, where the width of the fin was 150 mm at an angle of 30° to the horizontal. The shading applied is shown in Figures 8 and 9.

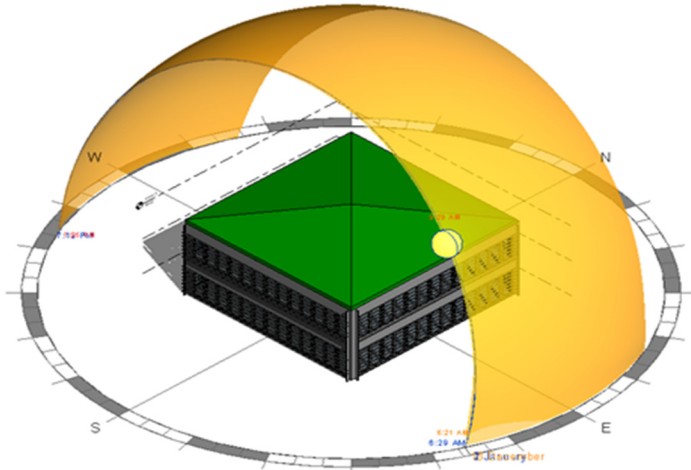

**Figure 8.** The simulation of the sun path and the effect of applied 150 mm horizontal fins with a 30° angle.

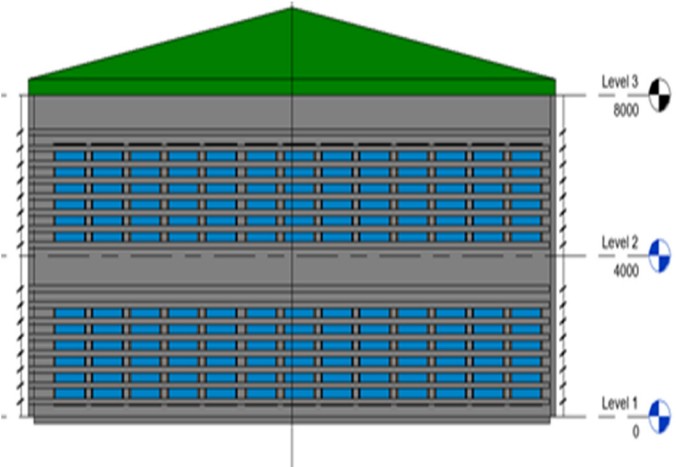

**Figure 9.** Frontal view simulation of the applied 150 mm horizontal fins with a 30° angle.

Then, we observed the effect of an overhang on the shading of the window, where the overhang had a length of 1100 mm from the edge of the wall; the overhang shading can be seen in Figure 10.

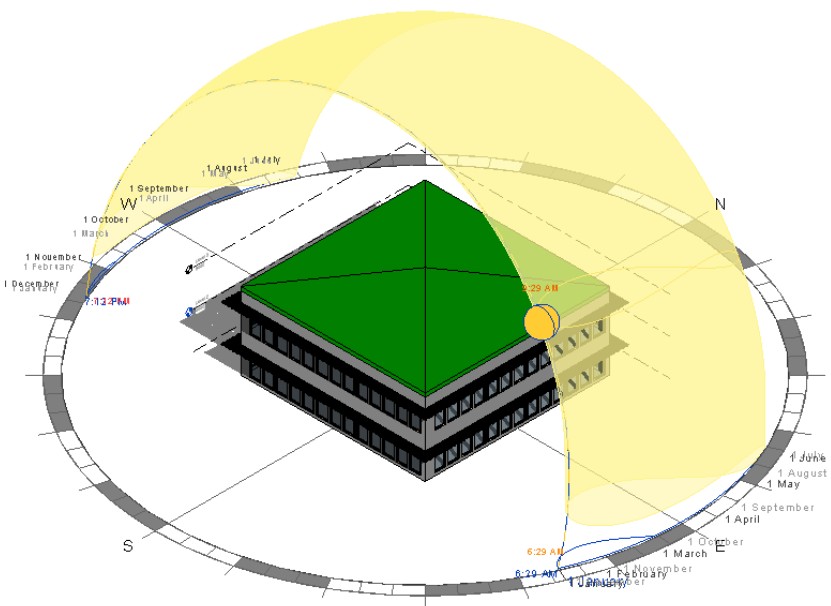

**Figure 10.** The simulation of the sun path and the effect of the applied 1100 mm overhang.

We then changed the length of the overhang to 1500 mm. The frontal view of the overhang can be seen in Figure 11, and we can see it in the form of the line as it is the front view.

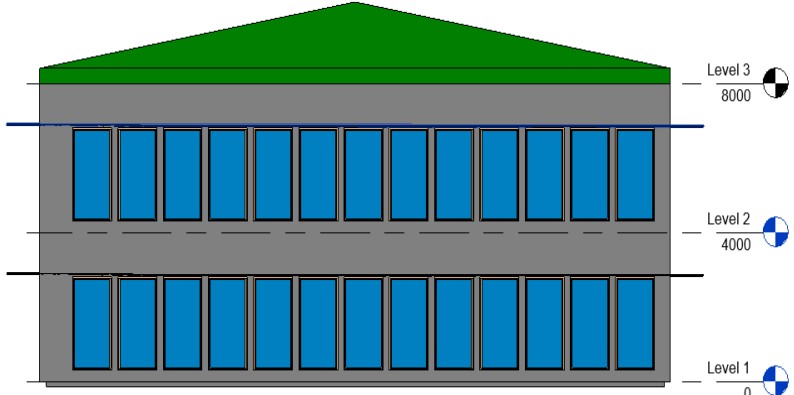

**Figure 11.** Frontal view simulation of the applied 1800 mm overhang shading.

In the next shading type, similar to a simple overhang louvre twisted along its horizontal axis, the shade provided different shading at different times of the day. One end provided more shade than the other. It also helped to add aesthetic value to the structure of the building and attracted more attention due to its unique design. Figures 12 and 13 show different views of the shading design applied to the building.

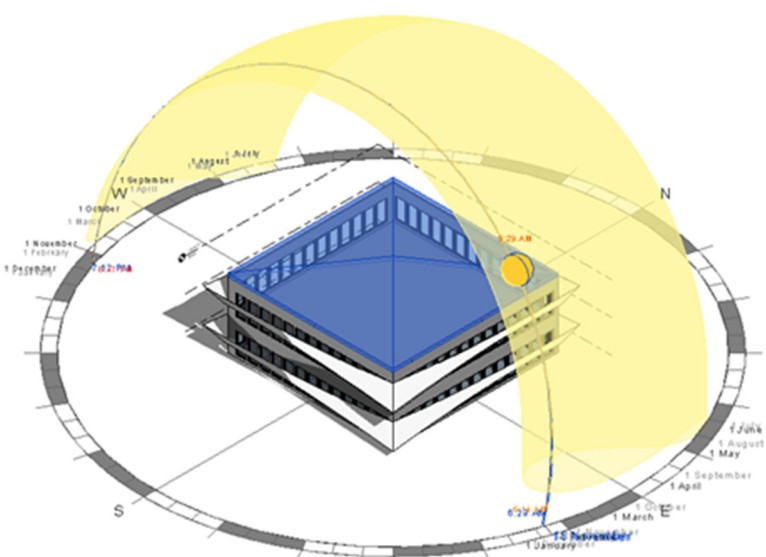

**Figure 12.** The simulation of the sun path and the effect of shading type applied.

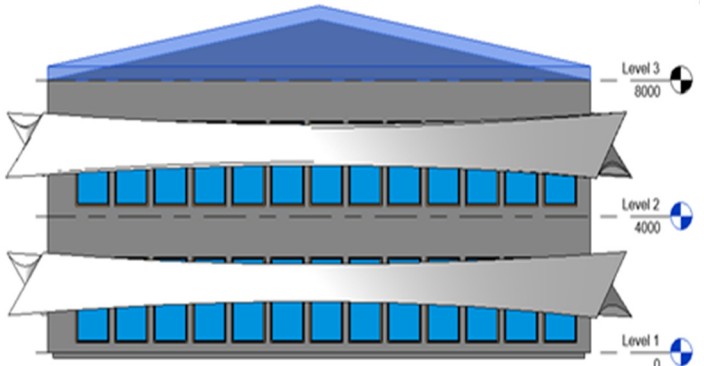

**Figure 13.** Frontal view simulation of the twisted horizontal shading applied.

The next type of shading was a modified version of the overhang with an inclination of 45°. The inclined overhang provided more shade to the window and for an extended period during the day, as shown in Figure 14. The length of the overhang was 1500 mm.

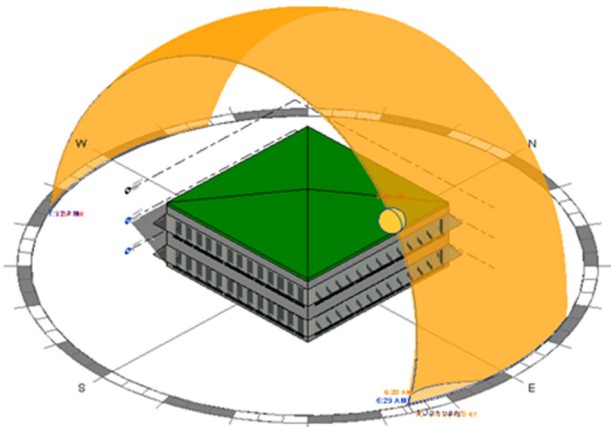

**Figure 14.** The simulation of the sun path and the effect of the applied horizontal, inclined louvres.

The cooling loads obtained for each type of shading applied are listed in Table 4.

**Table 4.** Cooling loads for various types of shadings.

| Shading | Cooling Load in kW |
|---|---|
| Without shading | 175.15 |
| 8 horizontal fins 0° (250 mm) | 162.35 |
| 8 horizontal fins 0° (150 mm) | 165.58 |
| 8 horizontal fins 30° (150 mm) | 161.28 |
| Horizontal louvres overhang (1000 mm) | 158.89 |
| Horizontal louvres overhang (1500 mm) | 153.45 |
| Twisted horizontal louvres 180° to 360° (1500 mm) | 155.96 |
| Horizontal louvres overhang inclined 45° (1500 mm) | 147.92 |

## 4. Results and Discussion

All simulations were carried out and a cooling load value was obtained for every shading type applied. The peak cooling load of the building resulted when no external shading was used on the building, giving a total requirement of 175.15 kW. When different shading types were added at the windows' level, other values were recorded, as shown in Table 4. These cooling loads were then compared with the building's original value of the cooling load, and energy savings were calculated, as shown in Table 5. A graphical representation of the cooling loads calculated can be seen in Figure 15.

**Table 5.** Energy savings achieved by different shading types.

| | Shading | Energy Saving (%) |
|---|---|---|
| 1. | Without shading | 0 |
| 2. | 8 horizontal fins 0° (250 mm) | 7.308 |
| 3. | 8 horizontal fins 0° (150 mm) | 5.464 |
| 4. | 8 horizontal fins 30° (150 mm) | 7.919 |
| 5. | Horizontal louvres overhang (1000 mm) | 9.283 |
| 6. | Horizontal louvres overhang (1500 mm) | 12.389 |
| 7. | Twisted horizontal louvres 180° to 360° (1500 mm) | 10.956 |
| 8. | Horizontal louvres overhang inclined 45° (1500 mm) | 15.547 |

Note: It is essential to mention that the energy savings as a percentage of energy saved by applying the shading were calculated separately and individually due to the core differences of shading types that were simulated. Therefore, the total energy/consumptions of the building were not investigated because of the abovementioned reason.

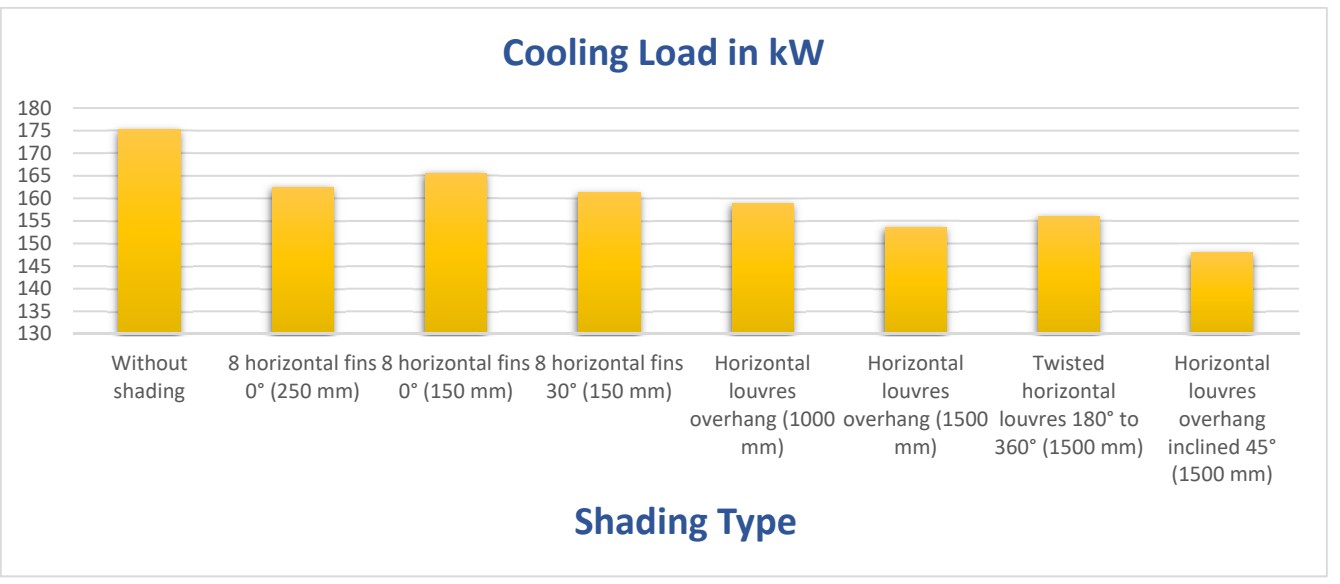

**Figure 15.** Graphical representation of the energy saving due to the effect of different shading types.

One of the major concerns is the installation and material costs of the shading type when selecting the style that best fits the building. From the first three types, observation showed that the horizontal fins of 250 mm would use more materials than those with 150 mm fins. Apart from that, horizontal fins with 0° angles acted very similar to the overhang on top of the window. Still, apart from that, horizontal fins with a 0° angle worked identically to the overhang on top of the window, except that the overhangs were applied throughout the window height; therefore, a greater area of the window was shaded to provided a good amount of shade to the window. Moreover, the lighting quality in the building was also good and did not impact the visual comfort levels of the occupants. More savings were observed when longer louvres were used, but the results were similar to the smaller louvres, except they were tilted at the angle of 30° to the window. Therefore, instead of using longer louvres, shorter louvres can be used with a slightly tilted angle to cause the same effect as the longer louvres

The percentage of energy saved by applying each shading type was also calculated. This comparison was made with reference to the scenario of no shading used. Table 5 shows the energy savings as a percentage relative to the original building.

When the overhang shades with lengths of 1000 mm and 1500 mm were applied to the building, the calculated cooling loads observed showed that the longer the overhang length, the lower the cooling load. It helped to reduce the unwanted radiation reaching the window. The longer overhang provided shade most of the time during summer. It confirmed the basics of the overhang shade design. The shadows provided overall energy savings of 9 and 12% for shorter and longer overhangs, respectively.

Twisted single louvre was used, and the cooling loads calculated provided an energy saving of almost 11%. This high energy saving was also due to the immense size of the shading, which was 1500 mm. The last shading type applied was a single overhang inclined at 45°, providing the maximum energy saving of 15.54%. Inclining by 45° provided shade throughout the day for an extended period and blocked rays from reaching directly to the window at a much higher rate. The energy savings trend as a percentage relative to no shading used to the building for all the shading types applied.

## 5. Conclusions

In conclusion, different shadings can be applied to buildings to reduce the cooling loads. Revit software was successfully used to better understand the additional analyses needed to be done before a building is approved. It was shown that changing the geometry and size of the shading can impact the overall performance of the building.

It was found that a single overhang provided the best results at an angle of 45° with a length of 1500 mm. The straight overhang was less beneficial than the inclined one. Inclination helped with blocking more rays for an extended time of the day, thus impacting the overall performance of the building. However, the increased length of the shade demanded a more robust structure of the building, especially when considering very high-rise buildings and windy areas.

Consequently, external shading devices minimise the cooling required of a building, resulting in energy savings. The shade's efficiency is determined by the building's form, the shading design, and the amount and inclination of glazing. Furthermore, heat is obtained and wasted from existing dwellings as a result of air leakage or draughts. To avoid draughts at home, seal gaps around doors and windows in the houses, as this will significantly reduce the energy and lower costs.

Furthermore, shading devices will reduce the cost of cleaning the external façade, as the maintenance costs will be lower compared to the traditional methods that are currently used. In addition, the shading devices will contribute to reducing the electricity consumption, which will also lead to a decrease in the cost of electricity in general and the cooling system equipment specifically. From the above discussion, it can be concluded that this overall cost-effective saving of the shading devices will create a new economic valuation based on these technological tools that were examined. Additionally, it is very essential

to mention that the percentages of energy saved by applying the shading were calculated separately and individually due to the core differences of shading types that were simulated. Therefore, the total energy/consumptions of the building were not investigated because of the abovementioned reason; this could be done in further investigation. Moreover, since the peak loads were only calculated, this presents a limitation of this study; future work should calculate both the peak loads and energy consumptions and identify the relationship between them.

In the final analysis, this study contributed to the development of knowledge regarding the optimisation of shading devices in buildings by considering the daylight admission, particularly in the Darwin Casuarina town hall building, regarding the context of a tropical region.

**Author Contributions:** Conceptualisation, A.M.; Data curation, A.M., H.M.-R. and A.W.M.N.; Resources, A.W.M.N. and Z.Z.; Investigation.; A.M., M.A.U.R.T., H.M.-R. and S.S.; Project administration, A.M., M.A.U.R.T., A.W.M.N. and S.S.; Writing—original draft preparation, Z.Z., H.M.-R. and M.A.U.R.T.; Writing—review and editing, A.M., M.A.U.R.T., S.S., H.M.-R., M.A.U.R.T. and M.M. All authors have read and agreed to the published version of the manuscript.

**Funding:** This research received no external funding.

**Institutional Review Board Statement:** Not applicable.

**Informed Consent Statement:** Not applicable.

**Data Availability Statement:** Not applicable.

**Conflicts of Interest:** The authors declare no conflict of interest.

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
