# Peer review of "Reducing the Cooling Loads of Buildings Using Shading Devices: A Case Study in Darwin"

_sustainability, doi:10.3390/su14073775_

Round 1

Reviewer 1 Report

Dear authors,

after a careful reading I suggest to strengthen and deepen the proposed research. I found out some lack in the methodogy, but the greater problem is the absence of any innovation in the methodology and in the results provided.

Methodology.

Fig. 1 is unclear. It is described as a flowchart, but it doesn't refer to a flowchart lexicon. How does the left tree relate with the right one? What is the direction of the "flow"?

I sincerely didn't find any innovation in methodology. If you perform a scientific paper, you should provide robust reasons about why, and for whom, the work is interesting, with scientific soundness and carrying a progress in research. 

Why don't you refer to an optimization strategy / information management tool to individuate the best solution, possibly in integration with other key requirements in sustainability subjected to the effects of shading devices?

Results.

Results are limited only to . If managed results confirm your hypothesis, why don't you integrate it with daylighting evaluaitons, or economic valuation among those technological solutions you examined?

Please provide an English polishing check, some expression seems to be part of a colloquial language.

Minor revisions.

Abstract:

  • is it correct to describe the intensity of solar irradiation as "high number of solar radiations"?

Introduction:

  • please check how reference [24] has been introducted in the text. Maybe it is a typo due to reference manager.

Materials and methods:

  • please correct the in-text reference to Fig. 2;
  • where is Fig. 4, cited in Section 2.1?
  • verify reference to Table 2, currently missing.

Reviewer 2 Report

The submitted paper is a numerical work based on computer simulation aimed at identifying potential energy savings in buildings based on the use of different shading devices in hot climate areas. It does so by modelling a case study building located in Darwin, Australia and applying a series of shading solutions to it to quantify the potential energy saved and identify the best one among those.

Although the structure of the paper is clear and the methodology is well defined, a few key issue remain that needs to be addressed. Following comments are separated into 2 sections, major comments and minor ones.

Major comments:

  1. The use of English throughout the paper should be improved, some examples are included in the minor comments, but are not exhaustive.
  2. The study focuses on the use of shading devices, however introduction and literature review discuss multiple aspects of energy losses in buildings, including insulation materials and PCMs, the link between those aspects and the content of the paper is unclear
  3. Additionally, no literature review is provided on other studies related to shading devices, and how they compare with the current work being presented. This should be included
  4. Although the applied methodology is clear, not enough details are given about the simulation software used (HAP), is it a dynamic model? How are solar radiation and and shading devices modelled in it? is reflection from shading modelled in the tool? More details necessary
  5. The different shading options should be more clearly defined, through the use of numbering or other reference, and should always be reported in the same order for improved clarity, Figure 5 through 15 could be subsequently formatted differently to be more useful
  6. Throughout the paper there is a lack of understanding of what is the output of the simulation, since loads and energy are mentioned, while the results reported in Table 5 seem to suggest only peak loads are being compared, and therefore the total energy/consumptions of the building are not investigated. This needs to be clarified and confusion between loads, energy and consumption avoided.
  7. If the simulations preformed only calculate Peak loads, some attention should be given to the relation between peak loads and energy consumptions, and if/how the limitations of this study limit any conclusion on the overall energy savings of the building.

Minor comments:

  1. The use of the word "tremendously" in the abstract should be reconsidered
  2. statement about how building components are responsible for high cooling leads should be contextualized within the nature of the work (hot climate),a s other aspects become dominant in other climates
  3. Page 2 par.3 "plane" instead of "plan"?
  4. The use of the word However throughout the paper should be reconsidered
  5. Bottom of page 4 words missing before [24]
  6. "Windows contributes to most of the heat gain/loss in buildings." at page 3 is an overgeneralized statement and should be more specific.
  7. When referring to the case study, at least the country should always be included, given the international nature of the journal
  8. "Darwin was selected as the shading because of its experiences with an adequate amount of sunny and partly sunny days. Direct sunlight can be blocked or reduced to the window, which is the weakest link for thermal losses occurring in a building." this statement at the bottom of page 4 is very confusing.
  9. Typos in last paragraph of page 5
  10. Typo in Assumptions on page 6 instead of "consumptions"?
  11. Unclear assumption on how internal loads are treated, since it mention both being ignored and "maintained"
  12. "Mean Daily Peak" in Table 1 needs to be clarified, as not enough details are provided
  13. Error in first paragraph of page 7
  14. Use uniform number of decimals in table 2
  15. Figure 15 and 16 are reporting the same results, also the shading options should be included in the same order as the rest of the paper.

Reviewer 3 Report

The paper presents the issue of reducing the cooling loads of buildings using optimised shading devices on an Australian case study. Researches made by Authors were justified and clearly described.

The review revealed the need for two major and general comments:

  1. Did the authors take into account the differentiation of the shades in the S, E-W and N directions? The authors pay attention to the reduction of building materials, and the article does not indicate whether the reduction of the curtains on the northern facade of the building would be justified.
  2. Is it possible to install PV panels as sun shades? How could this affect the building's energy balance? I believe that the authors should pay attention to this possibility in the article.

In general paper is well written, but I also have dozen minor suggestions for Authors to consider:

  1. Page 4 –  last paragraph, should recall figure 2 instead of 1
  2. Page 5 –  Figure 3 numeric values are unreadable, this also applies to drawings 5, 7, 9, 11, 13, 15
  3. Page 6 –  Steps in p.2.1 are hardly legible, could be written as numbered list
  4. Page 7 – improper reference to the publication “Error! Reference source not found..”
  5. Page 7 –  different types of shading could be more distinguishable. Could they be separated and named as e.g. a, b, c, etc? this would facilitate interpretability and order in the captions under the figures
  6. Page 11 –  Figure 15 is not recalled in text, without frontal view of this shading type
  7. Page 12 – table 4 with results shouldn’t be in p.3 named results? Cooling load in table 4 should be in kW unit
  8. Page 13 – doubled figure 15, this one has bad caption, cause it’s about energy savings
  9. Page 14 – figure 16 is redundant as it duplicates the content of figure 15

Reviewer 4 Report

The paper reports the paper reports a study on the use of shading to reduce cooling load on buildings base don a study in Darwin.  This is a topical subject area with need to reduce energy demand and environmental impacts The study presents some interesting information but lacks depth, with details lacking on the methodology applied and an in depth discussion on the results and the reasons for rationale behind underlying the data. The English/grammar contains numerous errors, please check manuscript.

Specific comments

Introduction

Abstract  

Solar radiations needs to be clearly defined as to what is meant by solar radiations

Introduction

Much of the introduction should be re-written to improve the English and hence improve the understanding of the reader of the review material provided.

50% of the population live in cities at present, it is predicted this will rise to 70%, not that the WHO predicts it will be 50%. This is an example of where the English and grammar need revising.

The authors should be given for reference 24, not simply [24].

Materials and Methods

The authors should provide the rationale for the selection of the building. It is assumed that this is a building in Darwin which has been fitted with sensors to monitor the solar radiation and energy levels?

Provide suitable references and a brief description of the software and the application in the context of the study. What is justification for selection, ie reference data that shows more accurate results.

Specify what changes in components are used and why this is justified. Give references to support statement that windows and shading devices responsible for most cooling load.

Provide justification as to why materials and equipment are ignored.

How is data collected. Give specifics on sensors used. Locations and frequency of monitoring to determine energy use

Provide data and references for angle measurements used in study.

As the objective sentence bottom page 5 is repeated.

Justification for assumption of Variable Volume Duct system. This assumption is repeated in sections 21 and 2.2.

What was the justification for the shading devices selected and the angle of the devices.

Figures 5, 7, 9, 11 and 15, are similar why are they included, please identify the differences in each figure to show the unique details in each figure. Also suggest a more detailed figure for the different shading orientations 8, 8,10 and 12 to clearly illustrate the orientation, length etc to demonstrate the differences.

Results and Discussion

Is the cooling load determine dover 1 year? State length of study explicitly in the paper and how the data is obtained, ie sensors or model, if model what parameters are used in the model.

Authors state the lighting quality was good. How was this measured?  What is a good amount of shading, how was this judged to be good?

Authors state that the long overhang provided shade most of the time, what is actual time of the daylight hours that shading is provided. This data should be included in paper how much unwanted radiation did it prevent reaching the windows?

Round 2

Reviewer 1 Report

Dear authors,

I appreciated all the specifications you provided about study scope and limitations.

I think that the the Figures 1 and 3 now help to better understand the effects of different shading devices in the considered context. Results discussion benefits of this clarity.

Please carefully check out guidelines for final manuscript production, in terms of figures and captions placement, blank lines, and misleading font colours.

Author Response

Thank you for supporting the paper publication and pointing out some minor formatting issues.  The authors have rechecked the manuscript and have addressed all remaining formatting issues such as figures and captions placement, blank lines, and misleading font colours. The revised manuscript is free from such errors, now.

Reviewer 2 Report

All the previous comments have been addressed, and no further comments are needed. Some minor improvements to the English used in the paper might be needed before publication, especially in the added paragraphs.

The work is interesting and well presented.

Author Response

Thank you for supporting the publication and suggesting language proofreading. The authors have improved the English in the manuscript now and especially in the added paragraphs.
